# The NLRP3 Inflammasome: Metabolic Regulation and Contribution to Inflammaging

**DOI:** 10.3390/cells9081808

**Published:** 2020-07-30

**Authors:** Allison K. Meyers, Xuewei Zhu

**Affiliations:** 1Department of Microbiology and Immunology, Wake Forest School of Medicine, Winston-Salem, NC 27157, USA; ameyers@wakehealth.edu; 2Department of Internal Medicine, Section of Molecular Medicine, Wake Forest School of Medicine, Winston-Salem, NC 27157, USA

**Keywords:** NLRP3 inflammasome, mitochondria, metabolism, inflammation, aging

## Abstract

In response to inflammatory stimuli, immune cells reconfigure their metabolism and bioenergetics to generate energy and substrates for cell survival and to launch immune effector functions. As a critical component of the innate immune system, the nucleotide-binding and oligomerization domain, leucine-rich repeat, and pyrin domain-containing 3 (NLRP3) inflammasome can be activated by various endogenous and exogenous danger signals. Activation of this cytosolic multiprotein complex triggers the release of the pro-inflammatory cytokines interleukin (IL)-1β and IL-18 and initiates pyroptosis, an inflammatory form of programmed cell death. The NLRP3 inflammasome fuels both chronic and acute inflammatory conditions and is critical in the emergence of inflammaging. Recent advances have highlighted that various metabolic pathways converge as potent regulators of the NLRP3 inflammasome. This review focuses on our current understanding of the metabolic regulation of the NLRP3 inflammasome activation, and the contribution of the NLRP3 inflammasome to inflammaging.

## 1. Introduction

Inflammation is the immune response to a variety of exogenously derived pathogen-associated molecular patterns (PAMPs) and endogenously derived danger-associated molecular patterns (DAMPs). Because sources of inflammation are so diverse and can lead to diseases that are both acute and chronic, it is critical to gain a mechanistic understanding of how inflammation is regulated. Furthermore, both chronic and acute inflammatory disorders affect all stages of life but, notably, converge and worsen in aged individuals. Therefore, an understanding of how inflammation is regulated is essential for promoting health with the ultimate goal of healthy aging.

The innate immune system, including neutrophils, monocytes, and macrophages, provides the first line of host defense against pathogens [1]. These cells sense and respond to PAMPs and DAMPs through the use of pattern-recognition receptors such as Toll-like receptors (TLRs) and nucleotide-binding and oligomerization domain (NOD)-like receptors (NLRs) [2,3,4]. As a member of the NLR family, inflammasomes are a group of large cytosolic multimeric protein complexes that process and cleave pro-inflammatory cytokines, such as pro-IL-1β and pro-IL-18, into a mature form [5,6,7,8,9,10]. Following maturation, the cytokines IL-1β and IL-18 alert the immune system to potential danger and enhance a pro-inflammatory immune reaction [11].

In general, inflammasome complexes consist of sensors, adaptors, and effectors. There are two main subtypes of inflammasomes, distinguished by the sensor protein that initiates their signaling. They are either NOD and leucine-rich repeat (NLR) family proteins or absent in melanoma 2 (AIM2)- like receptors (ALR) family proteins [7,8]. Adaptor proteins allow for the association of the sensor and effector proteins, such as apoptosis-associated speck-like protein (ASC), which contains a caspase recruitment domain that can bind to several caspase effector proteins [12]. The NLRP3 inflammasome, as an example, is composed of the sensor NLRP3, the adaptor ASC, and the effector caspase-1. As the most well-characterized inflammasome, the NLRP3 inflammasome has been implicated in many different disease pathologies [8,13].

Activation of the NLRP3 inflammasome in macrophages occurs in two steps, each with a different activating signal [5,6,7,8,9,10]. First, macrophages are primed through the recognition of an initial “danger” signal (Signal 1), which induces the transcription and production of inactive pro-IL-1β and NLRP3, which is subsequently ubiquitinated. Typically, the recognition of the bacterial cell wall component lipopolysaccharide (LPS), a PAMP, by Toll-like receptor 4 (TLR4) acts as a priming signal for innate immune cells such as monocytes or macrophages and activates transcription via NF-κB. Second, the recognition of a second activation signal initiates assembly of the inflammasome complex (Signal 2). Second signals are notably diverse, such as mitochondrial oxidative damage, lysosomal membrane rupture, and plasma membrane potassium efflux [5,6,7,8,9,10]. Currently, how macrophages assemble the inflammasome complex in response to a variety of danger signals is still not entirely clear [9]. It is known that NLRP3 is de-ubiquitinated and associates with ASC, which then associates with pro-caspase-1, forming a large multimeric protein complex [8]. Pro-caspase-1 undergoes autoproteolytic cleavage, likely as a result of proximity-induced multimerization, and produces the active form of caspase-1 [7]. Active caspase-1 is an aspartate-specific cysteine protease that cleaves pro-IL-1β and pro-IL-18 into their mature forms. After maturation, IL-1β is rapidly secreted from cells to induce inflammation. Pyroptosis, a highly inflammatory form of programmed cell death, can also occur following NLRP3 inflammasome activation [14,15,16,17,18,19]. This response is likely to reduce the replication of intracellular pathogens but is also thought to contribute to immunosuppression in diseases such as sepsis [8].

Because of the diverse nature of NLRP3 inflammasome stimuli, many studies have aimed to understand the exact mechanisms that lead to NLRP3 inflammasome activation. These studies have revealed that the NLRP3 inflammasome is heavily regulated by cellular metabolism [6,20]. Mounting evidence suggests that cellular metabolism is a crucial driver of cellular function, such as the metabolic control of macrophage polarization and inflammation [21,22,23]. In this review, we will focus on the current understanding of how cellular metabolism regulates the activity of the NLRP3 inflammasome, and then discuss its implications on our understanding of inflammatory diseases and “inflammaging”.

## 2. Mitochondria as the Central Metabolic Organelle

Mitochondria, well known as the powerhouse of the cell, act as a critical regulator of many cellular processes such as cell death, cellular signaling, and energetic homeostasis [24]. Mitochondrial dynamics, such as number and location, profoundly influence the metabolic status of a cell [25]. Inflammasomes are highly tuned to this as well. Evidence has shown that the NLRP3 inflammasome utilizes several mitochondria centric mechanisms to assemble (Figure 1). Firstly, NLRP3 possesses an N-terminal sequence that allows for it to localize to the mitochondria. Upon activation, components of the NLRP3 inflammasome translocate to mitochondria, an event dependent by the adaptor protein mitochondrial antiviral signaling protein (MAVS) [26]. Microtubules also promote the localization of the NLRP3 inflammasome to the mitochondria in the presence of activating signals by mediating the association of ASC on the mitochondria to NLRP3 [27]. The association of ASC and NLRP3 was also shown to be facilitated by calcium flux [28]. Finally, this localization is facilitated by cardiolipin, a mitochondria specific phospholipid. Cardiolipin can translocate from the inner mitochondrial membrane to the outer mitochondrial membrane, where cardiolipin directly binds NLRP3 to promote its activation [29]. Notably, the abundance of cardiolipin is linked to several pathologies, such as Barth syndrome [30]. Further studies are required to determine whether altered cardiolipin abundance affects the activation of the NLRP3 inflammasome.

Mitochondrial DNA (mtDNA) is also shown to promote inflammasome activation (Figure 1). Mitochondrial damage following environmental or metabolic stress induces oxidation of mtDNA. Oxidized mtDNA can be released into the cytosol, where it binds NLRP3, leading to inflammasome activation and IL-1β secretion [24]. More recently, it has been demonstrated that specifically, newly synthesized mtDNA is critical for the NLRP3 inflammasome activation [31]. Mechanistically, LPS stimulation activates transcription factor interferon regulator factor (IRF)1. IRF1 then induces the transcription of deoxyribonucleotide kinase UMP-CMPK2 (CMPK2), which allows for mitochondrial DNA replication. The newly synthesized DNA is oxidized, released into the cytosol, and promotes the inflammasome complex assembly [31]. Together, these findings emphasize the central role that mitochondria play in the activation of the NLRP3 inflammasome, especially during the process of complex assembly.

### 2.1. Oxidative Phosphorylation and Citric Acid Cycle Metabolites

The production of ATP in the mitochondria via oxidative phosphorylation is a process intimately linked with the tricarboxylic acid (TCA) cycle. The TCA cycle produces NADH (nicotinamide adenine dinucleotide) molecules necessary for the donation of electrons to the electron transport chain (ETC). Additionally, the TCA cycle enzyme succinate dehydrogenase, which catalyzes the conversion of succinate to fumarate, is Complex II in the ETC. Enhanced glycolysis and the flux of glycolysis-derived carbons into the TCA are necessary for the activation of innate immune cells such as macrophages and dendritic cells [32,33]. Concurrently, acute immune activation suppresses oxidative phosphorylation-mediated ATP production and favors Warburg glycolysis, a state that is coupled with increased succinate levels [34]. Notably, both of these changes facilitate NLRP3 inflammasome activation (Figure 2). First, studies using bone marrow-derived macrophages indicate that inhibition of the ETC complex I with rotenone can induce NLRP3 inflammasome activation [35]. In this system, reactive oxygen species (ROS) production was dispensable for inflammasome activation. Instead, the NLRP3 inflammasome activation was more likely a consequence of mitochondrial damage from depolarization, suggesting that ETC failure acts to increase the sensitivity of macrophages to NLRP3 inflammasome stimuli [35]. Second, high levels of the metabolite succinate can support IL-1β expression by stabilizing hypoxia-inducible factor 1-alpha (HIF-1α) for IL-1 β transcript expression to occur [36]. This process has recently shown to be inhibited by the anti-inflammatory metabolite itaconate [37], a TCA cycle off-shoot metabolite that is produced by decarboxylation of cis-aconitate of the TCA cycle by the enzyme immune responsive gene 1. Itaconate inhibits the activity of succinate dehydrogenase and mitochondrial respiration in LPS-activated macrophages, concurrently, decreasing LPS-induced mtROS [37,38,39]. Itaconate can also activate the anti-inflammatory cellular programming of nuclear factor erythroid 2-related factor 2 (Nrf2) via alkylation of cysteine residues on the protein KEAP1 [40]. Additionally, itaconate can inhibit LPS-induced IL-1β secretion by impairing glycolytic flux via targeting glycolytic enzymes GAPDH or fructose-bisphosphate aldolase A [41,42]. Collectively, these studies highlight the importance of the ETC and the cycling of the TCA cycle in the regulation of the NLRP3 inflammasome (Figure 2).

### 2.2. Cell Stress and Redox Regulation

Cellular stress is notably diverse and can have dramatic implications on cellular function. In eukaryotic cells, ROS are generated by NADPH oxidases (NOXs) at the cell membrane and in multiple organelles, as well as via mitochondrial respiration and other metabolic processes [43,44]. Mitochondrial ROS is a potent NLRP3 inflammasome activator and is one indicator of cellular stress [45]. Diseases such as colitis and sepsis are exacerbated by redox imbalances, particularly high levels of ROS. In the case of colitis, one study [46] demonstrated that inflammasome activation could be inhibited via activation of Nrf2 and its downstream response protein heme oxygenase-1 (HO-1), responsible for intracellular antioxidant defenses (Figure 1). Consistent with this, treatment of macrophages with the tripeptide toosendanin increased Nrf2 signaling, which attenuated NLRP3 inflammasome activation, decreasing the polarization of macrophages towards the pro-inflammatory (M1) phenotype, thus ameliorating the pathological damages of colitis [46]. Another cellular redox system known to regulate the activation of the NLRP3 inflammasome is the thioredoxin-interacting protein (TXNIP) [47]. TXNIP is known to associate with the oxidant scavenging enzyme thioredoxin. When levels of mtROS are high, TXNIP dissociates from thioredoxin so it can scavenge oxygen radicals [48]. In inflammasome activation conditions, mtROS induces dissociation of TXNIP from thioredoxin. Free TXNIP then binds to NLRP3 to induce the activation of the inflammasome. Consistent with this, the synthetic flavonoid compound VI-16 has been shown to inactivate the NLRP3 inflammasome by reducing oxidative stress, which inhibits TXNIP from binding to NLRP3 [49]. Moreover, high levels of inflammasome activation markers were associated with sepsis-induced myocardial injury, along with increased levels of TXNIP [50]. Inhibition of TXNIP was shown to decrease ROS production, attenuate apoptosis, and reduce IL-1β secretion [50]. This study suggests that TXNIP/NLRP3 signaling is critical for the pathogenesis of sepsis-induced myocardial damage (Figure 1). Lastly, transient receptor potential melastatin 2 (TRPM2), an oxidative stress-sensitive calcium channel, was implicated in the regulation of the NLRP3 inflammasome [51]. Mechanistically, mitochondrial ROS induces calcium influx in a TRPM2 manner, which is crucial for IL-1β secretion.

Autophagy is a cellular self-cleaning process through which damaged organelles, such as mitochondria, and other aggregated products, are broken down into nutrients [52,53]. Autophagy is critical to maintaining cellular metabolic balance, not only for the supply of metabolites, but importantly, it regulates mitochondrial turnover [54]. Mitochondria are the location for several key metabolic processes, and as such, maintenance of proper mitochondrial biology and function is necessary for cells to survive and function. Unsurprisingly, mitochondrial damage has been linked to many diseases, and autophagy deficiency exaggerates mitochondrial injuries [55]. Autophagy inactivates inflammasomes through multiple mechanisms (Figure 1). For example, autophagy can target pro-IL-1β and ubiquitinated inflammasome complex for destruction [56,57]. Autophagy can also remove damaged mitochondria and suppress the release of mtROS and mtDNA [58,59]. Abnormal autophagy or mitophagy (the selective autophagic degradation of mitochondria) has been implicated in the aberrant activation of the NLRP3 inflammasome in metabolic diseases such as atherosclerosis and type 2 diabetes [60,61]. Interestingly, the NLRP3 inflammasome and autophagy reciprocally regulate each other and this relationship was, in part, demonstrated to be spatial in nature such that the inflammasome and autophagosomes colocalize in a P62-dependent fashion [57]. P62, a protein adaptor for the autophagic protein LC3, binds to K63 linked polyubiquitination on inflammasome components to deliver inflammasomes to autophagosomes [57].

Mitochondrial health has been shown to be central for the inhibition of the inflammasome by autophagy, such that accumulation of damaged and high ROS generating mitochondria, coupled with low autophagic flux leads to inflammasome activation [59]. One early study shows that the depletion of the autophagic proteins LC3B and Beclin-1 enhances the activation of caspase-1 and IL-1β secretion, as a result of decreased mitochondrial homeostasis, increased production of ROS, and the release of mtDNA in an NLRP3-dependent manner [58]. Similarly, another study showed that loss of mitophagy by a deficiency in PINK1 or PARK2 induces ROS production in renal tubule epithelial cells, which activates the inflammasome and increases apoptosis [62]. These findings suggest that PINK1-Parkin-induced mitophagy protects against inflammation via inactivation of the NLRP3 inflammasome. It has been further documented that ATF4, a transcription factor involved in the endoplasmic reticulum (ER) stress, can upregulate Parkin expression, which then activates mitophagy and leads to inactivation of the NLRP3 inflammasome [63]. Lastly, the mitochondrial cellular stress response protein Sirtuin (SIRT)3 is implicated in the negative regulation of autophagy on the NLRP3 inflammasome. Overexpression of SIRT3 significantly reduced palmitate-induced NLRP3 inflammasome by deacetylating, thus activating the autophagic protein ATG5. ATG5 activation then promotes autophagic flux and attenuates mitochondrial oxidative stress [64].

Collectively, these studies demonstrate the central role of mitochondria in NLRP3 inflammasome activation. Furthermore, an imbalance in cellular redox metabolism and dysfunction of mitophagy drive the inflammasome activation via multiple mechanisms.

## 3. Lipids

Lipids are implicated as key regulators of macrophage function, especially in cases of aberrant activation of inflammatory pathways. Notably, growing evidence demonstrates the crucial function of various lipid species in the regulation of the NLRP3 inflammasome [65]. Here, we focus on fatty acids and cholesterol species and their roles in regulating the NLRP3 inflammasome (Figure 3). Other lipid species, such as phospholipids and lipid mediators, have been well discussed by several reviews [66,67,68].

### 3.1. Fatty Acids

Fatty acids contain long hydrocarbon chains with variable degrees of saturation and a carboxyl group at one end. These molecules are used as components of other lipids, and play critical roles in biological membranes, energy homeostasis, protein localization, and act as signaling molecules. The heterogeneity of these lipids allows for them to interact with the NLRP3 inflammasome as both positive and negative regulators.

#### 3.1.1. Fatty Acid Metabolism

Lipids yield more energy than carbohydrates upon catalysis, and as such, their regulation is essential for maintaining a proper energy balance within the cell. Unsurprisingly, the mirrored activities of fatty acid synthesis and fatty acid degradation have both been implicated in the regulation of the NLRP3 inflammasome (Figure 3). It has been shown that the mitochondrial uncoupling protein (UCP-2) and fatty acid synthase (FASN), a key lipid synthesis enzyme, are critical regulators of inflammation [69]. UCP-2 is involved in regulating glucose metabolism and the expression of lipid regulatory enzymes [70]. In response to sepsis challenge or LPS, macrophages increase UCP-2 expression. Increased UCP-2 upregulates FASN expression, leading to increased macrophage lipid synthesis, which enhanced transcription of NLRP3 and IL-1β via the Akt and P38 MAPK pathways [69].

Interestingly, fatty acid oxidation (FAO) also can activate the NLRP3 inflammasome via NOX4 [71]. Mechanistically, NOX4 enhances the expression of carnitine palmitoyltransferase 1A (CPT1A), a key enzyme in the FAO pathway. Inhibition of NOX4- or CPT1A-mediated FAO inactivates the NLRP3 inflammasome, suggesting that NOX4-dependent activation of FAO via CPT1A is critical for the NLRP3 inflammasome activation. Though this study did not establish the direct mechanism by which CPT1A activity activates the NLRP3 inflammasome, the authors hypothesize that AMP-activated protein kinase (AMPK) regulation could potentially play a role in this activation [72]. ROS likely acts as the link between FAO and NLRP3 inflammasome activation as NOX4-derived mtROS is critical for the regulation of CPT1A protein expression [71].

These seemingly conflicting findings, being that both degradation and synthesis of fatty acids activate the inflammasome, likely indicate that metabolic imbalance itself and not necessarily the specific pathways act as a cue to activate an inflammatory response. Alternatively, these pathways could work through a similar mediator, such as mtROS, which is known to activate the NLRP3 inflammasome directly. Further research needs to be conducted for a better understanding of this regulation.

#### 3.1.2. Fatty Acid Saturation

As mentioned, fatty acids can have highly variable degrees of carbon chain saturation, which dictates their function. Most studies implicate two specific fatty acids, the saturated fatty acid, palmitic acid, and the monounsaturated fatty acid, oleic acid, in the regulation of the inflammasome, both of these lipids can be synthesized in humans, and they can also be taken in by diet (Figure 3). Saturated fatty acid-enriched high-fat diets are well known to be associated with inflammation and, as such, are increasingly being linked to inflammasome activity [60,73,74,75,76,77,78]. For example, it has been shown that exogenous palmitate inactivates AMPK, which impairs autophagy and promotes the generation of mtROS [60]. The production of mtROS then drives the NLRP3 inflammasome activation and IL-1β secretion.

In addition to activation of the NLRP3 inflammasome, palmitic acid has also been shown to cause many cellular perturbations such as ER stress, decreased cellular viability, loss of mitochondrial membrane potential, and the induction of many inflammatory genes such as IL-6 and TNF [79,80]. Mechanistically, palmitic acid and other saturated fatty acids can be crystallized in macrophages, which causes lysosomal membrane rupture, a known NLRP3 inflammasome trigger [76]. In contrast, oleic acid inhibits and even mitigates the effects of palmitic acid, such that the treatment of macrophages with oleic acid inhibits IL-1β release in a dose-dependent manner [76]. Additionally, in HepG2 cells, oleic acid is able to decrease NLRP3 inflammasome activation and pyroptosis by reducing ER stress to counteract the effects of palmitic acid [80].

Long-chain (≥20 carbons) polyunsaturated fatty acids (PUFAs) also regulate NLRP3 inflammasome activation (Figure 3). The ω-3 PUFA, docosahexaenoic acid (DHA), is a potent inhibitor of both caspase-1 activation and IL-1β secretion [81,82]. In addition to DHA, other ω-3 PUFAs, such as eicosapentaenoic acid and α-linolenic acid, but not ω-6 fatty acids, such as dihomo-γ-linolenic acid, can also inhibit inflammasome activation [81]. ω-3 PUFAs activate the surface receptors G-protein coupled receptors (GPR)120 and GPR40. Activation of the GPRs leads to the association of β-arrestin 2 with GPRs, followed by internalization of β-arrestin 2, which then binds to NLRP3 and blocks the NLRP3 inflammasome assembly and activation. It is worth noting that this study showed that oleic acid was unable to confer inhibition, as other studies discussed previously could [81]. Similarly, ω-3 PUFAs inhibit TLR4 signaling via activation of GPR120, which blocks the binding TAB1(TAK1 binding protein)/TAK1 (transforming growth factor-β activated kinase 1) in a β-arrestin2/TAB1-dependent manner [83]. Taken together, ω-3 PUFAs inactivate the NLRP3 inflammasome by dampening both priming and inflammasome assembly via the cell-surface receptor GPR120 and GPR40. In our study, dietary supplementation of ω-3 PUFAs, as well as ω-6 PUFAs, suppressed NLRP3 inflammasome activation by enhancing autophagic flux [78]. Together, these studies indicate that lipid saturation is critical in either activation or inhibition of the NLRP3 inflammasome, and further research is needed to distinguish how lipid saturation can accomplish these activities.

### 3.2. Cholesterol

Cholesterol is another type of lipid with a wide array of functions. It is taken in by diet but can also be synthesized in cells. High cholesterol leads to atherosclerosis, a condition of an imbalance in lipid metabolism and immune responses driven by the accumulation of cholesterol-laden macrophages in arteries [84,85,86]. The NLRP3 inflammasome has been implicated in the pathogenesis and progression of atherosclerosis, especially in potentiating sterile inflammation.

Oxidized low-density lipoprotein (oxLDL), a cholesterol carrier that promotes atherosclerosis, can form crystals within lysosomes after internalization by macrophages (Figure 3). Crystalized cholesterol disrupts the lysosomal membrane, which activates the NLRP3 inflammasome [87]. CD36, a scavenger receptor on macrophages, mediates endocytosis of oxLDL and is necessary for this activation [88]. More studies since the publication of these findings have gone on to demonstrate the mechanisms of cholesterol-induced activation of the NLRP3 inflammasome. Firstly, oxLDL was shown to activate NF-κB, which allowed for the upregulation of NLRP3, a pathway that is suppressed by the activity of fibronectin domain-containing protein 5 (FNDC5) [89]. FNDC5 is a membrane protein that has previously been shown to ameliorate metabolic perturbations during high-fat diet via AMP- dependent autophagy [90]. FNDC5 suppresses the NLRP3 inflammasome by blocking NF-κB activation in an AMPK-dependent manner [89]. Additionally, AMPK activates mitophagy and cholesterol efflux, which further inhibit NLRP3 inflammasome assembly [91].

Additional studies have demonstrated that treatment with oxLDL causes decreased cholesterol efflux from cells via the ATP binding cassette A1 (ABCA1), increased cathepsin B activity following lysosomal disruption, induction of ER stress, mitochondrial damage and increased ROS, and also increased potassium efflux. All of these effects were shown to contribute to the activation of the NLRP3 inflammasome [92,93,94]. Conversely, inhibition of the NLRP3 inflammasome was shown to decrease the formation of cholesterol-laden foamy macrophages [93,95,96]. These studies highlight the importance of metabolic homeostasis in inflammation, such that even in sterile conditions, the NLRP3 inflammasome can be activated and produce very robust responses.

## 4. Carbohydrates

### 4.1. Glycolysis

It is well characterized that macrophages undergo metabolic reprogramming to sustain their functions, and it is widely accepted that pro-inflammatory macrophages rapidly increase their rate of glycolysis [67]. Glycolysis is the metabolic process in which glucose is catabolized into pyruvate through a series of enzymatic reactions. Because of its proven role in macrophage activation and the central role of its products in other cellular processes, glycolysis is considered to be a major regulator of the NLRP3 inflammasome.

#### 4.1.1. Glycolytic Flux

To date, it is still unclear whether glycolytic flux positively or negatively regulates NLRP3 inflammasome activity. Some studies show that the inhibition of glycolytic flux attenuates the inflammasome. For example, one study showed that treatment of classically activated macrophages with aminooxyacetic acid, an inhibitor of aspartate aminotransferase, decreases glycolysis, concurrently reducing NLRP3 inflammasome activation [97]. This study suggests that glycolysis may promote the NLRP3 inflammasome. Consistently, another study also suggests that inhibition of glycolysis attenuates NLRP3 inflammasome activation, as shown by the use of the glycolysis inhibitor 2-deoxy-D-glucose (2-DG) [98]. 2-DG attenuated oxidative stress, which likely contributes to the NLRP3 inflammasome attenuation [98]. Hypoxic conditions, which are known to induce glycolysis, can activate the NLRP3 inflammasome [99,100]. Interestingly, glucose levels appear to regulate hypoxia-mediated NLRP3 activation such that a high glucose level in hypoxic conditions can promote glycolysis but is unable to activate the NLRP3 inflammasome [101]. This was shown to be the result of decreased cellular ATP levels, which were attributed to the opening of potassium efflux channels, a known NLRP3 inflammasome activator [101]. These studies suggest that, in part, glycolytic flux positively regulates the NLRP3 inflammasome (Figure 2).

Seemingly in conflict with these observations, a study demonstrated that the activation of the NLRP3 inflammasome in macrophages is inversely related to glycolysis such that inhibition of glycolysis activates the NLRP3 inflammasome [102]. It was shown that inhibition of glycolysis induces the NLRP3 inflammasome and pyroptosis as a result of increased mitochondrial ROS and perturbed NAD^+^/NADH ratios. Supplementation of downstream glycolytic metabolites in glycolytically impaired cells prevented inflammasome activation. It is worth noting that this study found that not all inflammasome activators are able to disrupt glycolytic flux, e.g., nigericin, and that disruption of glycolysis does not always activate the NLRP3 inflammasome, e.g., via 2-DG as other studies have found as well [102].

Further investigations are needed to understand this discrepancy as to whether glycolysis promotes or attenuates NLRP3 inflammasome activation. Nonetheless, cellular conditions such as cellular redox balance and metabolite prevalence may play a broader role in this regulation. Finally, with much attention being given to the flux of glucose through the glycolysis pathway as fuel for innate immune cell activation, evidence suggests that this is not wholly representative. Studies in dendritic cells demonstrate that early activation preferentially relies on glycogen metabolism to fuel rapid induction of glycolysis, whereas later activation relies on glucose utilization [103]. This observation was expanded to show that NLRP3 inflammasome priming and IL-1β secretion in dendritic cells is dependent on glycogen catabolism [104]. Interestingly, it has been shown recently that glycogen metabolism in macrophages produces an intermediate metabolite UDP-glucose, which activates STAT1-mediated inflammatory signaling transduction via the P2Y_14_ receptor [105]. Whether glycogen metabolism can also activate the NLRP3 inflammasome in macrophages is still unclear.

#### 4.1.2. Glycolytic Regulators

With this discrepancy in mind, several glycolysis regulators have been implicated in controlling the activity of the NLRP3 inflammasome (Figure 2). Hexokinase, a glycolytic enzyme that adds a phosphate group onto glucose to initiate glycolysis, was shown to be essential for NLRP3 inflammasome activation by activating glycolytic flux [106]. Furthermore, this activity is dependent on the mammalian target of rapamycin complex 1 (mTORC1). Deficiency of mTORC1 suppresses hexokinase expression and subsequently decreases glycolysis under inflammasome activating conditions [106]. Conversely, hexokinase was shown to act as a sensor for the Gram-negative bacterial product N-acetyl glucosamine. The sensation of N-acetyl glucosamine leads to hexokinase dissociation from the mitochondria, which inhibited hexokinase activity. Inhibition of hexokinase activity is shown to be sufficient to activate the NLRP3 inflammasome, possibly as a result of cytosolic mtDNA, which was also shown to increase [107]. Another well-known regulator is pyruvate kinase M2 (PKM2), which is responsible for catalyzing the final rate-limiting step of glycolysis. One study found that inflammasome activation is dependent on the activity of PKM2. IL-1β secretion resulted from PKM2-dependent lactate production, which promotes phosphorylation of eukaryotic translation initiation factor 2 alpha kinase 2 (EIF2AK2), a protein that is known to be implicated in inflammasome activation [108]. Beyond its role in the generation of pyruvate, PKM2 has also been shown to interact with HIF-1α to potentiate and sustain inflammasome activation [109]. Lastly, the tyrosine-protein kinase, Syk, was found to be a critical activator of the NLRP3 inflammasome in a study of the fungal immune stimulator, dectin. Dectin activates glycolysis by activating Syk via PI3K and Akt signaling, which is required for IL-1β secretion [110]. Collectively, these studies indicate that glycolysis is an essential element in the NLRP3 inflammasome system. Disruption of the glycolysis pathway broadly impacts NLRP3 inflammasome activity.

## 5. Other Metabolic Pathways

Besides the pathways discussed already, many other cellular processes are critical for cellular functions and can regulate the activity of the NLRP3 inflammasome.

### 5.1. Amino Acid Metabolism

The amino acid glutamate has previously been demonstrated to increase during ischemic conditions of glucose deprivation and hypoxia, which induced ER stress and TXNIP expression concurrent with IL-1β release [111]. Consistent with this finding, another study reported that glutamate activates the NLRP3 inflammasome via induction of ER oxidative stress [112]. Additionally, glutamate induces the influx of calcium and disrupts mitochondrial potential, which could be reversed using a synthetic antioxidative agent [112].

Arginine, via the enzyme protein arginine deiminase (PAD), has a previously undefined role in inflammasome assembly [113]. The PAD enzyme catalyzes the conversion of protein-bound arginine into citrulline, a posttranslational modification with the potential of altering protein activity. NLRP3 inflammasome priming and secondary stimuli induce citrullination synergistically via PAD2 and PAD4 isoforms, in a calcium-dependent manner. Inhibition of the PAD isoforms decreases IL-1β release and ASC aggregation but does not significantly alter the production of pro-IL-1β [113], suggesting that protein citrullination mainly impacts inflammasome assembly.

Lastly, the amino acid glycine was shown to inhibit NLRP3 inflammasome activation in a model of lung injury [114]. Mechanistically, exogenous glycine was found to attenuate LPS induced NF-κB activation, without altering protein abundance of the upstream proteins TLR4 or MyD88. Concurrently, exogenous glycine increased Nrf2 signaling, activating cellular oxidative stress responses such as HO-1. Together, exogenous glycine attenuates IL-1β production by decreasing the expression of NLRP3 inflammasome components [114]. Collectively, these studies demonstrate that the diverse pool of cellular amino acids has the ability to positively and negatively regulate the NLRP3 inflammasome (Figure 2).

### 5.2. Nucleotide Metabolism

ATP, the nucleotide containing DAMP, is one of the best-characterized stimuli of the NLRP3 inflammasome [115,116,117]. The ATP constituent adenosine has also been shown to regulate the inflammasome in a manner beyond the initial upregulation of IL-1β by NF-κB [118]. Mechanistically, adenosine amplifies signal 1 and signal 2 pathways by upregulating pro-IL-1β transcription via the cAMP-PKA (protein kinase A)-CREB (cAMP response element-binding protein)-HIF-1α pathway, in an adenosine receptor A_2A_-dependent manner [118].

Other nucleotide pathway metabolites have also been implicated in the regulation of the NLRP3 inflammasome (Figure 2). For example, the enzyme xanthine oxidoreductase (XOR), which catalyzes the conversion of hypoxanthine to xanthine or xanthine to uric acid during purine metabolism, has been shown to play a role in the activation of the NLRP3 inflammasome [119]. Inhibition of XOR attenuates IL-1β secretion partially by decreasing mtROS. Furthermore, in a mtROS-independent mechanism, inhibition of XOR restores mitochondrial bioenergetics via activation of the purine salvage pathway while reducing uric acid levels. Additionally, both soluble and crystalized uric acid stimulates the NLRP3 inflammasome in a MyD88- and/or mtROS-dependent manner, leading to the secretion of IL-1β [59,120,121,122]. Taken together, these studies demonstrate that nucleotide pathway metabolites are effective activators of NLRP3 inflammasome.

### 5.3. Acetylation and NAD Cycling

Acetylation is a posttranslational modification that acts to regulate protein activity. This modification has been linked to cellular metabolic status by the regulation of acetyltransferases and deacetylases by various metabolites. Most notably, NAD^+^-dependent deacetylases can sense the energy status of cells [123,124]. Increasing evidence suggests that NAD^+^ and acetylation are potent regulators of inflammasome activity (Figure 2). NLRP3 itself can be acetylated at Lys24 by the acetyltransferase KAT5 [125]. This acetylation event occurs during the assembly process to promote the NLRP3 inflammasome assembly. Disruption of NLRP3 acetylation blocks inflammasome activity. Additionally, acetylation of the cytoskeleton component α-tubulin mediates the assembly of the NLRP3 inflammasome [27]. Mechanistically, stimulation of the NLRP3 inflammasome diminishes cytosolic NAD^+^, which in turn inactivates the NAD^+^-dependent α-tubulin deacetylase SIRT2 [126], resulting in the accumulation of acetylated α-tubulin. Acetylated α-tubulin mediates dynein-dependent mitochondrial transport, which brings ASC on mitochondria in association with NLRP3 on the ER for complex assembly. Therefore, SIRT2 acts as a negative regulator of NLRP3 inflammasome activity [27]. Consistent with this study, other groups also reported that blockade of acetylation or restoration of intracellular NAD^+^ levels with various pharmaceuticals abrogates this phenomenon [127,128]. SIRT2 also removes acetyl groups from NLRP3 to block the complex assembly [129]. Despite this, a discrepancy with this mechanism exists. Evidence shows that elevated levels of intracellular NAD^+^ enhance IL-1β secretion. Additionally, this study found that sirtuin inhibition did not suppress NLRP3 inflammasome activity, and low levels of NAD^+^ were found to inhibit IL-1β secretion [130]. Further investigation into this discrepancy is needed. Nonetheless, it is clear that acetylation and NAD^+^ play a role in the activation of the NLRP3 inflammasome.

## 6. NLRP3 Inflammasome and Inflammaging

Despite the intended protective role of the NLRP3 inflammasome, and of inflammation in general, inappropriate or excessive inflammatory activity has been linked to disease pathology of both acute and chronic conditions. Chronic inflammation is usually considered to be low levels of long-lasting inflammation [131]. However, how inflammation begins and how exactly it contributes to various diseases is still not fully understood. Akin to the inflammation of chronic inflammatory conditions, inflammaging is the long term, low-grade immune activation from sterile sources that develops with aging. This type of inflammation, along with inflammation from metabolic sources termed metaflammation, contribute to the aging process [132], following a lifetime of inflammation.

Two well-characterized conditions, in which the aberrant activity of the NLRP3 inflammasome contributes to and exacerbates pathology, leading to inflammaging, are obesity and diabetes. Obesity and specifically obesity-associated insulin resistance have been linked to the activity of the NLRP3 inflammasome [133,134]. Saturated fatty acids such as palmitate, which are enriched in the high-fat diet, activate the NLRP3 inflammasome, driving insulin resistance [60]. NLRP3-dependent IL-1β secretion has been shown to impair pancreatic beta cell function [133,134,135], adipocyte function, and insulin sensitivity [136], promoting the progression of obesity and insulin resistance. The deletion of NLRP3 or inhibition of caspase-1 in mice was shown to improve insulin sensitivity and ameliorate obesity-associated pathologies [60,136,137]. Moreover, obesity accelerates age-related thymic atrophy and decreases T cell diversity [138]. Elimination of NLRP3 or ASC attenuates this age-related thymic atrophy and promotes T cell repertoire diversity [139]. In addition to the dysregulation of T cell homeostasis, age-associated B cell expansion in adipose tissues impairs tissue metabolism and promotes visceral adiposity in the elderly, a process regulated by the NLRP3 inflammasome [140]. Together, these two studies suggest that targeting the NLRP3 inflammasome has a potential beneficial effect on the re-establishment of immune competence in the elderly. Lastly, it was reported that individuals over 85 years of age could be stratified into two groups based on their expression level of inflammasome gene modules, as either constitutive or non-constitutive. The former group was associated with measures of all-cause mortality, again supporting the concept that targeting inflammasome components may ameliorate chronic inflammation and various other age-associated conditions [141]. Despite these findings, evidence also shows that IL-1β signaling is critical for acute islet compensation to metabolic stress and that IL-1 receptor type I deficient mice are more glucose intolerant than wile type and have impaired insulin secretory responses induced by high-fat-diet [142], suggesting the need for further study into the pathophysiological role of the NLRP3 inflammasome in acute and chronic metabolic conditions.

As discussed previously, atherosclerosis is another condition in which the activity of the NLRP3 inflammasome promotes pathogenesis. Atherogenic factors such as cholesterol crystals [87] and oxidized LDL [88] activate the NRLP3 inflammasome. Deletion or inhibition of the NLRP3 inflammasome complex, including NLRP3, ASC, or IL-1β, was shown to improve lipid metabolism, and to decrease inflammation, pyroptosis, and infiltration of more immune cells into plaques, thus ameliorating inflammatory responses and atherosclerosis progression [87,88,143,144,145,146,147,148,149,150,151]. Moreover, targeting the IL-1β pathway with canakinumab, a human anti-IL-1β antibody (a CANTOS clinical trial), at a dose of 150 mg every 3 months led to a significantly lower rate of recurrent cardiovascular events than placebo, independent of lipid-level lowering, further suggesting a beneficial effect of targeting the inflammasome with regard to cardiovascular outcomes [152]. Despite these findings, one study showed that atherosclerosis in apolipoprotein E-deficient mice is independent of the NRLP3 inflammasome [153]. Consistent with this study, another group reported that hematopoietic NLRP3 or caspase-1/11 deficiency had no effect on atherogenesis in LDL receptor-deficient mice [94]. The reason for these discrepancies is unknown and may result from different diet compositions, atherogenic mouse models used, or experimental conditions in different labs. Thus, the role of the NLRP3 inflammasome in the pathogenesis of atherosclerosis warrants further investigations.

Furthermore, a study using mice lacking NLRP3 inflammasome showed how impactful the NLRP3 inflammasome is in aged individuals, such that old mice without IL-1 receptor had improved measures of functional decline as compared to age-matched wild type [154]. This is consistent with the finding that activation of the NLRP3 inflammasome promotes systemic low-grade age-related inflammation in both the periphery and the brain that accelerates age-related degenerative changes [154]. Studies indicate that a lifetime of accumulating low-grade inflammation significantly alters gene expression profiles, thus adding a complicating factor to our understanding of any particular disease [132,155]. One example is the rheumatic disease osteoarthritis, a condition typically associated with aging. Osteoarthritis is intricately linked to the activity of the NLRP3 inflammasome [156,157,158] and to metabolism [159,160,161,162]. Another example is gout, an inflammatory condition within the joints driven by the precipitation of monosodium urate, a metabolite in the same pathway as xanthine oxidase [163].

Chronic NLRP3 inflammasome activity was shown to be exacerbated by defective autophagy in aged individuals, and enhancement of autophagy improves health outcomes [164]. Importantly, healthy metabolic status, such as the maintenance of SIRT2 activity, has been shown to have beneficial impacts on aging. Aging reduces SIRT2 expression and increases mitochondrial stress, leading to activation of the NLRP3 inflammasome in hematopoietic stem cells [165]. This age-driven functionality decline of hematopoietic stem cells was countered by SIRT2 overexpression, or NLRP3 inactivation [165]. Additionally, the activity of SIRT2 can prevent NLRP3 inflammasome-induced cell death [27]. In line with these findings, increased NAD^+^ levels along with sirtuin activation have been shown to improve mitochondrial homeostasis, organ function, and lifespan [166].

Complexing the matter of inflammaging is the observation that aged individuals have higher rates of sepsis incidence and mortality, resulting from the inability to launch an effective NLRP3 inflammasome response to infection [167,168,169]. Sepsis is one example of acute inflammation, being a severe infection that is short lived and usually presents with a considerably larger inflammatory response in comparison to chronic inflammation. Inflammasomal response is critical for host defense against the infection, such that mice lacking NLRP3 or ASC are more susceptible to bacterial infection [18,170]. On the other hand, excessive inflammasome activation, leading to exaggerated caspase-1 activation, IL-1β release, and pyroptosis, will cause severe cell and tissue damage and organ dysfunction, and ultimately death. So not surprisingly, patients with either too high or too low NLRP3 inflammasome signaling, as is the case for aged individuals, are potentially at higher risk of death [171]. Genetic or pharmacological inhibition of NLRP3 improves survival during polymicrobial sepsis or *E. coli* infection in mice [172,173]. Inactivation of the NLRP3 inflammasome by the manipulation of metabolic pathways, including glucose and cholesterol metabolism, also protects mice from sepsis [108,174]. Additionally, mice lacking NLRP3 also showed decreased lung injury as seen by decreased levels of epithelial death, immune infiltration, and pro-inflammatory cytokines, suggesting that the NLRP3 inflammasome exaggerates acute lung injury [175]. Together, these findings suggest that the presence of low-grade inflammation before secondary insult might negatively impact the abilities of an aged individual to respond to acute conditions, possibly due to the development of immune tolerance as indicated by inadequate cytokine responses. Further study is necessary to understand this phenomenon.

Together, these studies demonstrate how chronic and acute inflammatory and metabolic diseases can be exacerbated throughout the aging process, further complicating our understanding of the biology of the NLRP3 inflammasome and its role in the pathogenesis of diseases and aging.

## 7. Conclusions

Regulation of the innate immune sensor, the NLRP3 inflammasome, is fundamental for effective immune responses and in maintaining homeostasis. Cellular metabolites such as carbohydrates and lipids in their many forms can act as regulators of the NLRP3 inflammasome. Despite the certainty that metabolism does play a role in regulating the NLRP3 inflammasome, at present, some questions still remain, e.g., the definition of how NLRP3 directly senses metabolic perturbations, clarification on which lipids are positive and negative regulators, whether glycolysis is a positive or negative regulator of inflammasome activity, and how the activation of the NLRP3 inflammasome can, in turn, regulate cellular metabolism. Further investigation of these questions will help to advance our understanding of inflammasome activities and inflammasome-driven diseases and help to identify potential therapeutic targets of metabolic pathways and metabolites to prevent or treat inflammatory disorders.

## Figures and Tables

**Figure 1 cells-09-01808-f001:**
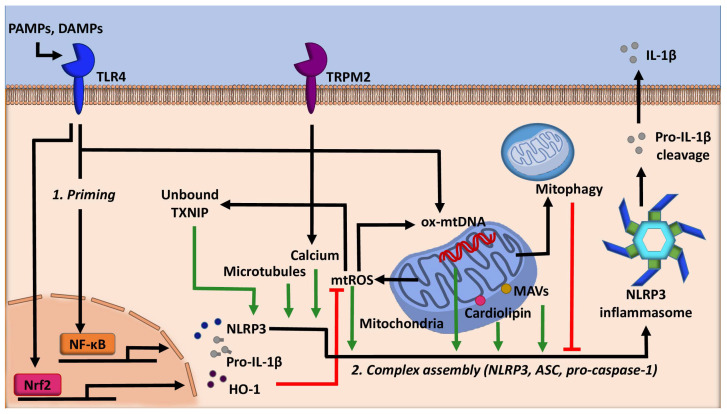
The role of mitochondria in regulating NLRP3 activity. Mitochondria are central in the regulation of the NLRP3 inflammasome. Mitochondrial damage leads to the production of mitochondrial reactive oxygen species (mtROS) and the release of newly synthesized mitochondrial DNA (mtDNA) into the cytosol, which gets oxidized (ox-), both stimulating the activation of the NLRP3 inflammasome. Priming signal activates nuclear factor erythroid 2-related factor 2 (Nrf2) and cellular stress responses such as heme-oxygenase-1 (HO-1), which mitigates the effects of mtROS and inhibit inflammasome activation. Mitophagy inhibits inflammasome activation by destructing pro-IL-1β and inflammasome complex, attenuating mitochondrial oxidative stress, and reducing mtDNA synthesis. mtROS stimulates thioredoxin-interacting protein (TXNIP) to release thioredoxin. Thioredoxin-free TXNIP then directly binds NLRP3 to enhance its activation. Additionally, microtubules, calcium signaling, cardiolipin, and mitochondrial antiviral-signaling protein (MAVs) aid in the assembly of NLRP3 inflammasome complex on the mitochondria, enhancing its activity. Green lines denote the promotion of NLRP3 activity, whereas red lines denote the inhibition of NLRP3 inflammasome activity. ASC: apoptosis-associated speck-like protein, DAMPs: danger-associated molecular patterns, PAMPs: pathogen-associated molecular patterns, TLR4: Toll-like receptor 4, and TRPM2: transient receptor potential melastatin 2.

**Figure 2 cells-09-01808-f002:**
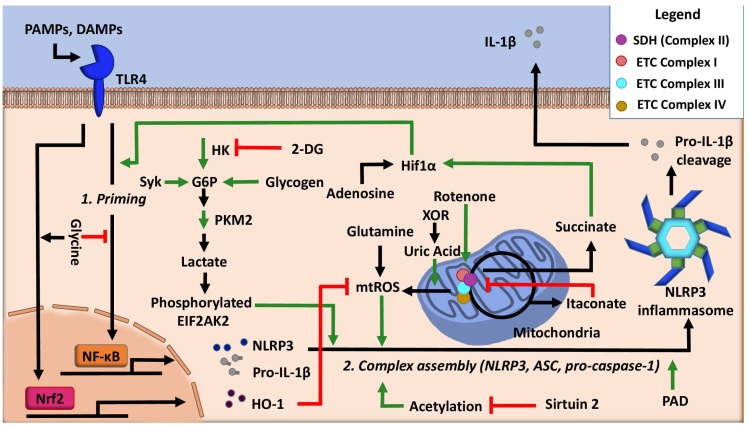
Regulation of NLRP3 inflammasome activity by cellular metabolic pathways. Stimulating macrophages with a priming signal increases the production of succinate, which stabilizes hypoxia-inducible factor 1-alpha (HIF-1α). Hif1α, also activated by adenosine, enhances the transcription of IL-1β. Inflammasome assembly is assisted by glycolytic flux and acetylation, which can be inhibited by 2-deoxy-D-glucose (2-DG) and Sirtuin 2, respectively. Itaconate inhibits succinate dehydrogenase (SDH) and reduces mitochondrial (mt)ROS, along with glycine, which enhances Nrf2-mediated cellular stress responses to inhibit inflammasome activation. Glutamine and xanthine oxidoreductase (XOR) via uric acid enhance mtROS. Green lines denote the promotion of NLRP3 activity, whereas red lines denote the inhibition of NLRP3 inflammasome activity. DAMPs: danger-associated molecular patterns, EIF2AK2: eukaryotic translation initiation factor 2 alpha kinase 2, G6P: glucose-6-phosphate, HK: hexokinase, PAD: protein arginine deiminase, PAMPs: pathogen-associated molecular patterns, and PKM2: pyruvate kinase M2.

**Figure 3 cells-09-01808-f003:**
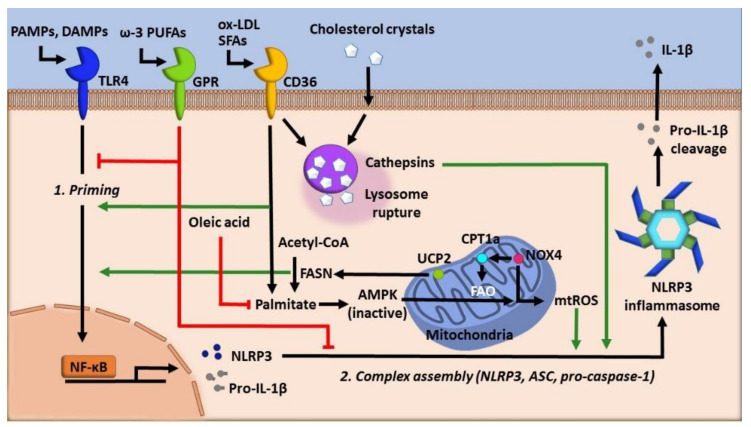
The role of lipids in regulating NLRP3 inflammasome. Lipid species are highly diverse and can either promote or inhibit NLRP3 inflammasome activity. Inflammasome priming, following the first hit, is promoted by both cholesterol crystals and the activity of fatty acid synthase (FASN) during fatty acid synthesis, whereas ω-3 polyunsaturated fatty acids (PUFA) such as docosahexaenoic acid (DHA) inhibit priming and inflammasome complex assembly via cell-surface receptors GPR120 and GPR40. Cholesterol crystals, oxidized LDL (ox-LDL), or saturated fatty acids (SFAs) can be internalized and form crystallized particles within lysosomes. Lipid crystallization can lead to lysosomal membrane rupture, which promotes NLRP3 activation in part by releasing cathepsins and enhancing mtROS production. The saturated fatty acid palmitate can also activate NLRP3 inflammasome by inactivating AMP-activated protein kinase (AMPK) and autophagy, leading to increased mtROS, whereas the monounsaturated oleic acid can inhibit this process. Finally, the fatty acid oxidation pathway promotes inflammasome activation via NADPH oxidase 4 (NOX4)-dependent upregulation of carnitine palmitoyltransferase 1a (CPT1a), likely through mtROS as well. Green lines denote the promotion of NLRP3 activity, whereas red lines denote the inhibition of NLRP3 inflammasome activity. DAMPs: danger-associated molecular patterns, FAO: fatty acid oxidation, GPR: G-protein coupled receptors, PAMPs: pathogen-associated molecular patterns, and UCP-2: mitochondrial uncoupling protein.

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
