# Peer review of "The NLRP3 Inflammasome: Metabolic Regulation and Contribution to Inflammaging"

_cells, 2020, doi:10.3390/cells9081808_

Round 1
Reviewer 1 Report
The review of Meyers and Zhu is summarizing metabolic regulation of the NLRP3 inflammasome, and, in a subsection the contribution of NLRP3 inflammasome in ”inflammaging”. The topic of this manuscript is interesting. However, I have some remarks and suggestions.
Comments:
While the term “inflammaging” first appears in the title, in the manuscript “inflammaging” is first mentioned in the very end. From my point of view this is misleading, and the authors should consider to rephrase the title of the manuscript.
Abstract: Please define all the used abbreviations.
Line 32-33: Please add a reference.
Line 37: Please change the names into “pro-IL-1β and pro-IL-18” and add a reference.
Line 52: Please add a reference. Moreover, please describe what LPS is and where it is coming from.
Line 56: There is a typing mistake “(ROS))”.
Line 57: Please add a reference.
Line 62: “…that cleaves pro-IL-1β and pro-IL-18”.
Line 65: Please add a reference.
Line 71: Please add a reference.
Line 147: Please add a reference.
Line 155 – 161: Please add references.
Line 163: What is the difference between autophagy and mitophagy? Please add a detailed explanation.
Line 164: Please add a reference.
Line 197: typing mistake “,.”.
Multiple references are missing. Please add references in lines: 215, 232, 238, 253, 269, 272, 275, 289, 291, 343, 346, 348, 354, 375, 401, 408, 412, 415, 420, 422, 424, 431, 444, 449, 452, 495, 502.
Line 358-359: Typing mistake “e.g.,.
Line 415-416: Style: Both sentences start with “together”.
Line 454: Please check the language.
Line 477-478: From my point of view this sentence is critical as for example in organ transplantation, chronic inflammation or chronic rejection is anything but “low levels”…
Reviewer 2 Report
Meyers and Zhu show in their review article a connection between cell metabolism and NLRP3 inflammasome activation and inflammaging development. The review is well written, brings a several good information and covers several points. In addition, the review is inserted in a very interesting topic.
I have one comment to the authors: It is well known that Nox4 predominantly produces H2O2. How does Nox4 contribute to the ROS production in mitochondria? Is it not the superoxide the main ROS produced by mitochondria? Is there any link between H2O2 and mitochondria producing ROS? The authors should give mode details on how Nox4 regulates ROS production in mitochondria.
Reviewer 3 Report
The process of "inflammaging" has been studied intensely in recent years. The present work addresses novel important aspects regarding the intricate mechanisms through which the NLRP3 inflammasome as well as other structures contribute to metabolic regulation and inflammaging. The main subject of the article is of interest to researchers. Moreover, the work is well written, well organized and scientifically sound.
Please find observations/suggestions below:
- Line 89: "barth syndrome" = "Barth syndrome";
- I believe it would be interesting to add a few mentions of rheumatic diseases (such as osteoarthritis) in which these mechanisms have been studied previously.
Reviewer 4 Report
Meyers and colleagues presented an interesting review describing the current understanding of the metabolic regulation of the NLRP3 inflammasome activation, and the contribution of the NLRP3 inflammasome to inflammaging. In general, this work can represent an advance in this topic. The manuscript is very well written and theorically well designed, however I have the following major concern regarding this work.
In the paragraph 6 the authors interestingly focus on the role of the NLRP3 inflammasome, and of inflammation in general, in the context of disease pathology of both acute and chronic conditions. On the other hand, the first paragraphs of the review are merely descriptive of the mechanisms regulating NLRP3 inflammasome activation, without a critical contextualization of those mechanisms in specific pathologies. I would suggest to add in each paragraph, i.e. for each metabolite class, specific references to inflammasome-driven diseases or inflammatory disorders where the reported metabolic alterations can been detected.
Minor revision:
Typos and spelling errors.
Line 16 “Inflamamsome”
Round 2
Reviewer 1 Report
Please make sure, that the figures appear at the right place in the text and all figures have figure legends.
Reviewer 4 Report
The authors have addressed the points raised in my previous review, and publication of the manuscript as is recommended.